

# The high expression of MTH1 and NUDT5 predict a poor survival and are associated with malignancy of esophageal squamous cell carcinoma

Jing-Jing Wang[1], Teng-Hui Liu[2], Jin Li[3], Dan-Ni Li[3], Xin-Yuan Tian[3], Qiu-Geng Ouyang[2] and Jian-Ping Cai[1,3]

[1] Peking University Fifth School of Clinical Medicine, Beijing Hospital, Beijing, China
[2] School of Pharmacy, Wenzhou Medical University, Wenzhou, Zhejiang, China
[3] The Key Laboratory of Geriatrics, Beijing Institute of Geriatrics, Beijing Hospital, National Center of Gerontology, National Health Commission, Institute of Geriatric Medicine, Chinese Academy of Medical Sciences, Beijing, China

Corresponding author
Jian-Ping Cai, caijp61@vip.sina.com

## ABSTRACT

**Background:** MTH1 and NUDT5 effectively degrade nucleotides containing 8-oxoguanine. MTH1 and NUDT5 have been linked to the malignancy of multiple cancers. However, their functions in tumor growth and metastasis in esophageal squamous carcinoma (ESCC) remain obscure. Our present study aims to explore their prognostic value in ESCC and investigate their function in MTH1 or NUDT5-knockout tumor cells.

**Methods:** MTH1 and NUDT5 protein expression in ESCC adjacent normal tissues and tumor tissues was examined by immunohistochemistry staining. Kaplan–Meier curves were used to assess the association between their expression and overall survival (OS) in ESCC patients. Univariate and Multivariate Cox regression analyses were generated to determine the correlation between these protein expression and OS of ESCC patients. Protein expression in ESCC cell lines were measured by Western blotting. To explore the potential effects of the MTH1 and NUDT5 protein in ESCC, cell models with MTH1 or NUDT5 depletion were established. CCK-8, cell cycle, Western blotting, migration and invasion assays were performed.

**Results:** Our present study demonstrated that the levels of MTH1 and NUDT5 were upregulated in ESCC cell lines and ESCC tissues, the expression of MTH1 and NUDT5 in ESCC tissues was significantly higher than in adjacent non-tumorous, and higher levels of MTH1 and NUDT5 predicted a worse prognosis in patients with ESCC. MTH1 and NUDT5 are novel biomarkers of the progression of ESCC and a poor prognosis. We also found for the first time that the high expression of NUDT5 independently predicted lower OS in patients with ESCC (hazard ratio (HR) 1.751; 95% confidence interval (CI) [1.056–2.903]; $p = 0.030$). In addition, the depletion of MTH1 and NUDT5 strongly suppressed the proliferation of ESCC cells and significantly delayed the G1 phase of the cell cycle. Furthermore, we found that MTH1 and NUDT5 silencing inhibited epithelial–mesenchymal transition mainly by the MAPK/MEK/ERK dependent pathway, which in turn significantly decreased the cell migration and invasion of ESCC cells. Our results suggested that the overexpression of MTH1 and NUDT5 is probably involved in the tumor development and poor prognosis of ESCC.

## INTRODUCTION

Reactive oxygen species (ROS) are free radicals or pleiotropic molecules that are the normal products of mitochondrial metabolism during the utilization of oxygen (*Starkov et al., 2004*; *Turrens, 2003*). Under homeostasis, complex enzymatic systems are responsible for eliminating these highly reactive molecules, and antioxidant substances are capable of inactivating excessive ROS in order to reduce their toxicity (*Li, Jia & Trush, 2016b*). The production of cellular oxidants has been linked to mutation as well as the modification of the gene expression, which may ultimately lead to diseases, such as cancer (*El-Kenawi & Ruffell, 2017*; *Ishikawa et al., 2008*; *Nakabeppu, 2014*; *Storz, 2017*). ROS can oxidize biological macromolecules, including nucleic acids, lipids, amino acids and proteins (*Waris & Ahsan, 2006*). One of the most prevalent DNA lesions induced by ROS is 8-oxo-7, 8-dihydro-2′-deoxyguanine (8-oxo-dG) which occurs approximately once per million guanine residues within the human genome (*Cheng et al., 1992*).

The nucleoside diphosphate linked to some other moiety X (NUDIX) hydrolase protein MutT, which was first discovered in *Escherichia coli*, was identified as a sanitizer of the oxidized dNTP pool (*Bessman, Frick & O'Handley, 1996*). In mammalian cells, MTH1, the most distinct NUDIX enzyme (NUDT1 or NUDIX hydrolase 1), degrades oxidized purine nucleotides to preserve the integrity of nucleic acid (*Cheng et al., 1992*; *Gad et al., 2014*; *Mo, Maki & Sekiguchi, 1992*). MTH1 degrades 2-OH-dATP, 8-oxoGTP and 8-oxo-dGTP but hardly acts on 8-oxoGDP or 8-oxodGDP (*Ishibashi, Hayakawa & Sekiguchi, 2003*). In addition to the enzymes that act primarily on 8-oxo-dGTP and 8-oxo-GTP, it was found that NUDIX hydrolase 5 (NUDT5 or NUDIX5), which was originally characterized as an enzyme that cleaves sugar phosphates, is capable of degrading 8-oxo-dGDP to its monophosphate form (*Arimori et al., 2011*; *Ishibashi, Hayakawa & Sekiguchi, 2003*; *Kamiya et al., 2009*).

Recent biological function analyses have suggested that NUDIX enzymes play a crucial role in the cell survival and cell cycle regulation (*Carreras-Puigvert et al., 2017*; *Iyama et al., 2010*; *Pickup et al., 2019*). In the common cancer cell lines A549 and MCF7, MTH1 was shown to be essential for the proliferation (*Carreras-Puigvert et al., 2017*). However, NUDT5 was found to be essential for the proliferation of SW480 but have a less marked effect on the proliferation of CCD841 and A549 cells (*Carreras-Puigvert et al., 2017*). The cancer cell lines also exhibited various types of cell cycle effects with different degrees of NUDIX enzyme depletion. In MCF-7 breast cancer cells treated with hydrogen peroxide, an increase in the MTH1 expression was observed, accompanied by G1/S phase cell cycle arrest. However, in HeLa cells with inhibition of NUDT5, the cell cycle was arrested in the G1 phase, and the number of cells in the S and G2/M phases decreased (*Carreras-Puigvert et al., 2017*). MTH1 and NUDT5 were reported to be associated with biological malignancy in various cancers (*Li et al., 2017*; *Wang et al., 2017b*). For patients with gastric cancer, the overexpression of MTH1 was correlated

closely with an increase in invasive depth and a decrease in the survival (*Duan et al., 2017*). MTH1 and NUDT5 were overexpressed in colorectal cancer (CRC) tissues relative to the adjacent normal tissues. In patients with CRC, an increase in MTH1 or NUDT5 expression indicated a decrease in OS after surgical resection (*Li et al., 2017*).

Accurate staging of cancers plays an important role for prognosis of patients, it also guides optimal patient treatment. American Joint Committee on Cancer Tumor Node and Metastasis (AJCC-TNM) provides description of anatomy and rules for clinical and pathologic classification for various cancers. It was one of the worldwide benchmark for evaluating the accurate stage of cancers and widely used to estimate prognosis and treatment results (*Fleming, 2001*). Esophageal cancer is the eighth-most common cancer, and among different subtypes of esophageal cancer, esophageal squamous carcinoma (ESCC) is one of main histologic types, accounting for >80% of all esophageal cancer cases worldwide (*Screening & Prevention Editorial, 2002*).

However, the role of MTH1 and NUDT5 in the growth and metastasis of ESCC is not clear. In the current research, we quantified the expression of MTH1 and NUDT5 in ESCC tissues and six types of cell lines. The association of MTH1 and NUDT5 protein levels with clinical parameters and survival data was evaluated among patients with ESCC. We then explored the molecular and functional mechanisms by which MTH1 and NUDT5 influence ESCC. We also depleted MTH1 and NUDT5 proteins using a lentiviral vector harboring an RNAi sequence targeting the MTH1 and NUDT5 genes in order to investigate their effects on ESCC cell proliferation, migration, invasion and the cell cycle. Finally, we explored the signaling pathways underlying ESCC alteration.

## MATERIALS AND METHODS

### Quantification of the MTH1 and NUDT5 expression by immunohistochemistry

Two commercial tissue microarrays (TMAs) consisting of 99 ESCC patients were purchased from Shanghai Outdo Biotech (ODCT; Shanghai, China). TMAs consisting of 81 paired tumor and adjacent normal specimens and 18 ESCC tumor tissues singly. The experience was approved by the Ethics Review committee of Shanghai Outdo Biotech (ODCT; Shanghai, China). The ethical permit number was YB M-05-02. We obtained clinical data of 94 patients who possessed complete overall survival information. The ESCC diagnosis was confirmed by a post-operative pathological examination. The patients involved in the study had undergone surgery from January 2006 to October 2008, follow-up had been conducted until July 2015. Median survival time of these patients was 16 months (range: 2–107 months).

Immunohistochemistry experiments were performed as previously described (*Li et al., 2017*). Antibodies against MTH1(ab187531) and NUDT5(ab129172) were purchased from Abcam (Cambridge, MA, USA). Normal goat serum (5%) was incubated with the negative control section as a negative control. Visualization was carried out with a Nikon Eclipse 80i microscope (Nikon, Tokyo, Japan).

Immunohistochemistry staining was evaluated by taking into account both the percentage of positive cells and the intensity of staining (*Li et al., 2017*). On the basis of the percentage of immunoreactive cells, the intensity category was scored from 0 to 4, as follows: 0 (negative), 1 (1–25%), 2 (26–50%), 3 (51–75%) and 4 (76–100%). Staining intensity of the specimen was graded from 0 to 3, as follows: 0 (negative), 1 (slightly stained), 2 (moderately stained) or 3 (heavily stained). The final score was then determined as the proportion of positive cells multiplied by the intensity category. Based on the final score, the expression of MTH1 and NUDT5 was classified as either low immunoreactivity (0–6) or high immunoreactivity (7–12).

## Cell lines and cell culture

Six types of human esophageal squamous cell lines were obtained from Beijing Beina Chuanglian Co., Ltd., (BNBIO; Beijing, China), these cell lines were KYSE30, KYSE50, KYSE70, KYSE140, KYSE450, KYSE520 and checked in the NCBI databases. The human normal fibroblast cell line WI-38 and human lung fibroblast IMR-90 cells were obtained from the American Type Culture Collection (ATCC; Rockville, MD, USA). KYSE30 and KYSE450 cells were cultured in EMBM media with 20% FBS. WI-38, IMR-90, KYSE50, KYSE70, KYSE140 and KYSE520 cells were cultured in DMEM media with 10% FBS. All cell lines were cultured in a humidified incubator set to 5% $CO_2$.

## RNA interference and transfection

The nontargeting control short hairpin RNA CON077 (shRNA) (5′-TTCTCCGAA CGTGT CACGT-3′), MTH1 shRNA sequences (5′-CGACGACAGCTACTGGTTT-3′) and NUDT5 shRNA sequences (5′-ATGGATCCTACTGGTAAA-3′) were inserted into the lentiviral vector GV248-puro cut by AgeI/EcoRI sites. Lentivirus LV-MTH1-RNAi, LV-NUDT5-RNAi or LV-CON077-RNAi containing MTH1, NUDT5 or control shRNA, respectively, was then produced by transient transfection in 293T cells.

For transfection, KYSE50 or KYSE70 cells were seeded into 24-well plates at 60–70% confluence. After 24 h, the cells were cultured with virus particles LV-MTH1-RNAi, LV-NUDT5-RNAi or LV-CON077-RNAi in the presence of Polybrene® (4 µg/mL) for 24–48 h. We then used puromycin (one µg/ml final concentration) to select positive clones after transfection. The stable colonies were amplified after 14 days' selection on puromycin.

## Western blotting

Western blotting experiments were performed as previously described (*Li et al., 2017*). Commercial antibodies used in this study were as follows: antibodies against MTH1 and NUDT5 were purchased from Abcam (Cambridge, MA, USA); mouse anti-E-cadherin, anti-N-cadherin and anti-Vimentin antibodies were from BD Biosciences (San Jose, CA, USA); rabbit anti-AKT, anti-p-AKT, anti-MEK, anti-p-MEK, anti-ERK and anti-p-ERK were from CST (Danvers, MA, USA); anti-MMP-7, anti-p16, anti-p21 and anti-p27 were from ABclonal (Boston, MA, USA); and anti-Tubulin mouse antibody was from ZSGB-BIO (Beijing, China). The experiment was repeated three times.

### The cell counting kit-8 proliferation assay

KYSE cells were plated at $3 \times 10^3$ cells per well into 96-well plates and then incubated at 37 °C in a $CO_2$ incubator for 0, 24, 48 and 72 h sequentially. The increase in cells was calculated by the CCK-8 assay according to the manufacturer's protocol. Subsequently, the reagent (10 μl) (Beyotime, Jiangsu, China) was added to particular well. The cells were then incubated at 37 °C for 1 h in the humidified incubator. Optical density (450 nm) was monitored. Three wells were assessed per group, with the results expressed as the means ± standard deviation (SD). Each experiment was repeated three times.

### Cell cycle assay

Cells were seeded in 6-well plates and then maintained with 5% $CO_2$ for 36–48 h at 37 °C. Harvested the cells and subjected to washing with phosphate-buffered saline (PBS) and then fixed at 4 °C overnight with 70% ethanol. Removed the ethanol and suspended the cells with PBS. The cells were finally resuspended in a solution of PI (50 μl/ml) and RNase1 (250 μg/ml) for 30 min in a dark environment at room temperature. The cell cycle was measured using a flow cytometer at 488 nm. The cell-cycle phase was detected with the BD FACSD via a software program to analyze the percentages of cells in G0/G1, S and G2/M phases based on the DNA content.

### Cell migration and invasion assay

Cell migration was performed using 6.5-mm transwell permeable supports with 8.0-μm pore polycarbonate membranes inserts. Cell invasion was performed with 6.5-mm transwell inserts with 8.0-μm pore polycarbonate membranes which were pre-coated with Matrigel (Costar, Corning Incorporated, NY, USA) for invasion. The cells were suspended with serum-free DMEM, and $8 \times 10^4$ cells were seeded into the upper chambers in 200 μl of DMEM with 1% FBS, while DMEM containing 20% FBS (700 μl) was added to the lower chamber. After culturing for 48 h, the cells on the upper surface of the membrane were removed, and those on the bottom of the membrane were fixed with 10% formalin for 30 min and then stained with 0.1% crystal violet for 4 h. Cells from 10 random microscopic fields/insert were counted in triplicate at ×100 magnification. Calculated the mean number of migrated or invaded cells.

### Statistical analyses

All statistical analyses were performed using the SPSS software program (SPSS Inc., Chicago, IL, USA). The means were compared between two groups using Student's $t$-test. Pearson' $\chi^2$ or Fisher's exact tests was used to calculate the correlation between MTH1 or NUDT5 and clinicopathological features of patients with ESCC. OS was calculated by the Kaplan–Meier analysis and compared between two groups with the log-rank test. Effects of ESCC clinicopathological variables and the expression of MTH1 or NUDT5 protein on the survival and prognosis of ESCC patients were assessed with the Cox proportional hazards regressions model. $P$ values of less than 0.05 were considered to indicate statistical significance.

## RESULTS

### The protein expression of MTH1 and NUDT5 in human ESCC tissues: the high expression of MTH1 and NUDT5 predicted a poor prognosis

Immunohistochemical analyses of TMAs verified that MTH1 and NUDT5 proteins were highly expressed in tumor tissues (Figs. 1A–1H). The clinical details are listed in Table 1. MTH1 and NUDT5 were poorly detected in adjacent normal tissues, as revealed by immunochemistry (Figs. 1A–1J). Tumor tissues varied in staining intensity (weak or strong) (Figs. 1A–1H). Representative images of immunohistochemical staining of MTH1 and NUDT5 proteins are shown in Figs. 1A–1J. On the basis of intensity category and immunostaining intensity, among the 94 ESCC specimens, 43 exhibited relatively high levels of MTH1 and 58 showed relatively high NUDT5 expression (Figs. 1K and 1L), but no high expression was found in the adjacent tissues.

We then determined the association between the protein expression of MTH1 and NUDT5 and the clinicopathological characteristics of 94 ESCC specimens (Table 1). Significant positive correlations were identified between the expression of MTH1 and the AJCC and T stages ($p = 0.040$ and $p = 0.048$, respectively), as well as between the expression of NUDT5 and the AJCC stage and lymph node invasion ($p = 0.001$ and $p = 0.002$, respectively) (Pearson' $\chi^2$ or Fisher's exact tests, $p < 0.05$). No other statistically significant associations between the protein expression and established prognostic factors (N stage, cell differentiation, vascular invasion, gender and age) were observed (Table 1).

The association between the MTH1 and NUDT5 protein expression and the OS rate of ESCC patients was evaluated using a Kaplan–Meier survival analysis with the log-rank test. As shown in Figs. 1M and 1N, the patients with low MTH1 ($p = 0.0311$) and NUDT5 ($p = 0.0014$) levels had longer OS values than those with high levels. The results of a univariate Cox regression analysis indicated that the following characteristics were significantly associated with a reduced OS: AJCC stage ($p < 0.001$), T stage ($p = 0.011$), N stage ($p = 0.002$), MTH1 expression ($p = 0.014$) and NUDT5 expression ($p = 0.003$) (Tables 2 and 3). Furthermore, a multivariate Cox regression analysis demonstrated that the high expression of NUDT5 independently predicted a low OS in patients with ESCC (hazard ratio (HR) 1.751; 95% confidence interval (CI) [1.056–2.903]; $p = 0.030$) (Table 3).

### The MTH1 and NUDT5 protein expression in human ESCC cell lines: silencing MTH1 and NUDT5 inhibited the proliferation of ESCC cell lines

To assess the expression of MTH1 and NUDT5 as a whole, we detected the levels of these two proteins by performing Western blotting in six human ESCC cell lines (KYSE30, KYSE50, KYSE70, KYSE140, KYSE450 and KYSE520), the human normal diploid fibroblast line WI38 and the normal human fibroblast model of senescence IMR90 (Fig. 2A). The data indicated that the normalized proteins level of MTH1 and NUDT5 in

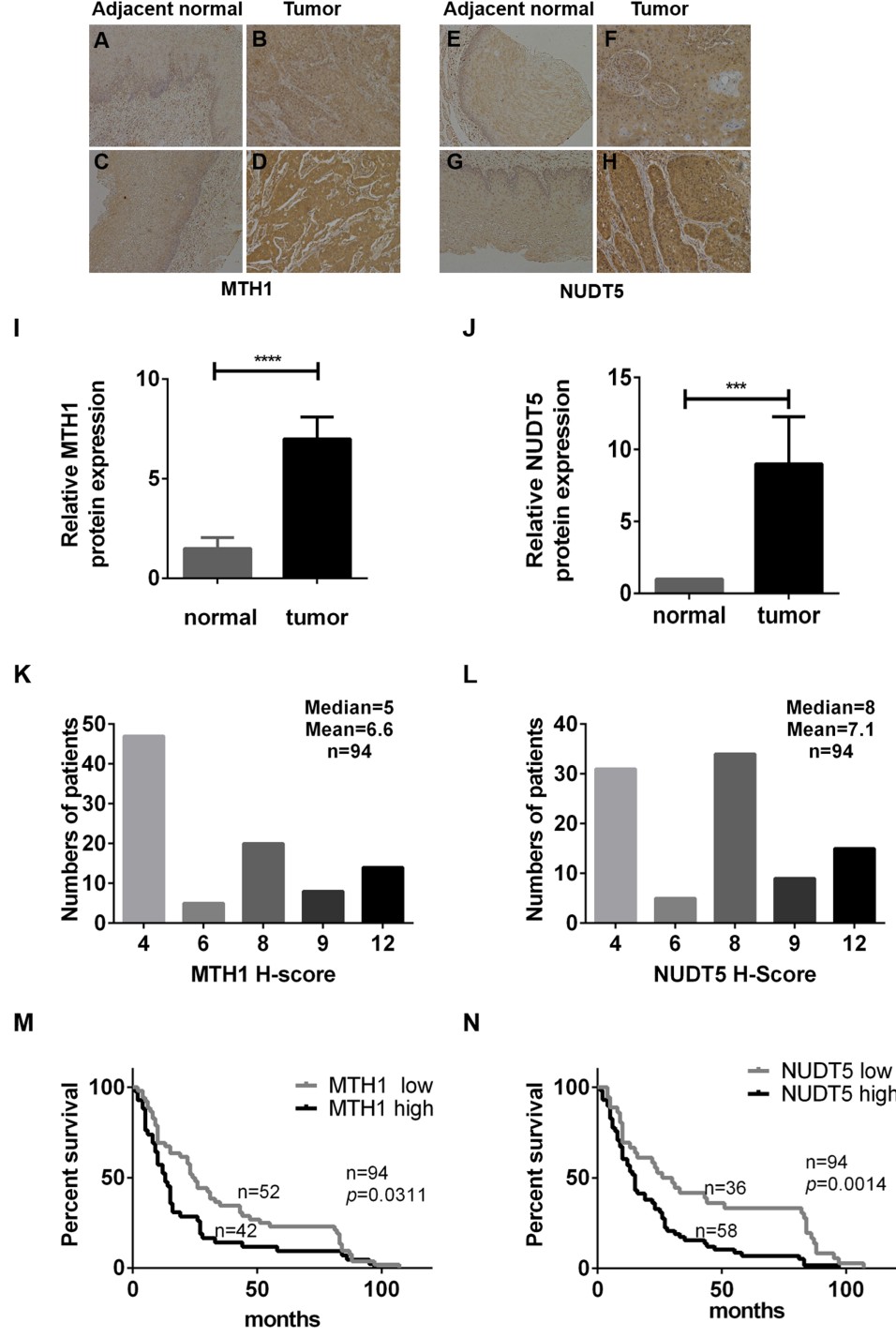

**Figure 1 The expression of MTH1 and NUDT5 protein in ESCC tissues and Kaplan–Meier curves for the overall survival rate of ESCC patients.** (A–N) The expression of MTH1 and NUDT5 protein in ESCC tissues was assessed by immunohistochemistry with anti-MTH1 or anti-NUDT5 antibodies. Representative immunohistochemical staining for the MTH1 and NUDT5 expression in ESCC specimens (X100) are shown in A–H. (A) Very low expression of MTH1 in adjacent normal tissue. (B) Low expression of MTH1 in tumor tissue. Tissues of (A) and (B) come from the same patient. (C) Very low expression of MTH1 in adjacent normal tissue. (D) High expression of MTH1 in tumor tissue. Tissues of (C) and (D) come from the same patient. (E) Very low NUDT5 expression in adjacent normal tissues.

**Figure 1 (continued)**
(F) Low expression of NUDT5 in tumor tissue. Tissues of (E) and (F) come from the same patient.
(G) Very low NUDT5 expression in adjacent normal tissue. (H) High expression of NUDT5 in tumor tissue. Tissues of (G) and (H) come from the same patient. (I) IHC score of MTH1 protein expression in adjacent normal tissue and tumor tissue of (A–D). (J) IHC score of NUDT5 protein expression in adjacent normal tissue and tumor tissue of (E–H) (Student's $t$-test, $^*p < 0.05$, compared with adjacent normal tissue). (K and L) Immunohistochemistry staining score distribution in ESCC patients. (M and N) Kaplan–Meier curves for the overall survival rate of ESCC patients. The association of the MTH1 and NUDT5 expression with the OS rate of ESCC patients was evaluated by a Kaplan–Meier survival analysis of immunohistochemical staining of TMAs. ESCC patients exhibiting a high expression of MTH1 and NUDT5 showed a lower survival than those with low levels ($p = 0.0311$ and $p = 0.0014$). $p$ values were calculated by the log-rank test.  

**Table 1 Relationship between the clinicopathological parameters and expression of MTH1 and NUDT5 proteins ($n = 94$).**

| | Total | MTH1 low | MTH1 high | P | Total | NUDT5 low | NUDT5 high | P |
|---|---|---|---|---|---|---|---|---|
| **Age (years)** | | | | | | | | |
| <65 | 48 | 30 | 18 | 0.213 | 48 | 20 | 28 | 0.972 |
| ≥65 | 46 | 22 | 24 | | 46 | 19 | 27 | |
| **Gender** | | | | | | | | |
| Male | 70 | 39 | 31 | 0.895 | 70 | 27 | 43 | 0.347 |
| Female | 24 | 13 | 11 | | 24 | 12 | 12 | |
| **AJCC stage** | | | | | | | | |
| I + II | 45 | 30 | 15 | 0.040* | 45 | 27 | 18 | 0.001* |
| III + IV | 49 | 22 | 27 | | 49 | 12 | 37 | |
| **T stage** | | | | | | | | |
| T1 + T2 | 15 | 12 | 3 | 0.048* | 15 | 9 | 6 | 0.154 |
| T3 + T4 | 79 | 40 | 39 | | 79 | 30 | 49 | |
| **N stage** | | | | | | | | |
| N0 | 42 | 28 | 14 | 0.061 | 80 | 32 | 48 | 0.562 |
| N1 + N2 | 52 | 24 | 28 | | 14 | 7 | 7 | |
| **Differentiation** | | | | | | | | |
| Well + Moderate | 67 | 41 | 26 | 0.108 | 67 | 28 | 39 | 0.139 |
| Poor | 27 | 11 | 16 | | 15 | 11 | 4 | |
| **Lymph node metastasis** | | | | | | | | |
| No | 80 | 41 | 39 | 0.081 | 42 | 25 | 17 | 0.002* |
| Yes | 14 | 11 | 3 | | 52 | 14 | 38 | |
| **Vascular invasion** | | | | | | | | |
| No | 90 | 50 | 40 | 1.000 | 90 | 39 | 51 | 1.000 |
| Yes | 4 | 2 | 2 | | 2 | 0 | 2 | |

**Notes:**
Fisher's exact test.
\* $P < 0.05$.

ESCC cell lines were significantly upregulated than those in the control cell lines WI38 and IMR90 (Student's $t$-test, $p < 0.05$) (Figs. 2B and 2C). All of these data showed that proteins level of MTH1 and NUDT5 were significantly upregulated in the ESCC cell lines.

**Table 2 Results of univariate and multivariate analyses of the overall survival of MTH1.**

| | Univariate analysis | | Multivariate analysis | |
|---|---|---|---|---|
| | HR (95% CI) | P | HR (95% CI) | P |
| Age (years) | | | | |
| <65 | 1 | | | |
| ≥65 | 0.892 [0.571–1.395] | 0.617 | | |
| Gender | | | | |
| Male | 1 | | | |
| Female | 0.604 [0.348–1.049] | 0.073 | | |
| Lymph node metastasis | | | | |
| No | 1 | | | |
| Yes | 1.377 [0.766–2.473] | 0.285 | | |
| AJCC stage | | | | |
| I + II | 1 | | 1 | |
| III + IV | 3.675 [1.896–7.126] | <0.001* | 1.940 [1.076–3.496] | 0.027* |
| T stage | | | | |
| T1 + T2 | 1 | | | |
| T3 + T4 | 2.615 [1.249–5.477] | 0.011* | 1.432 [0.622–3.296] | 0.398 |
| N stage | | | | |
| N0 | 1 | | 1 | |
| N1 + N2 | 2.065 [1.249–3.293] | 0.002* | 0.633 [0.211–1.899] | 0.415 |
| Differentiation | | | | |
| Well + Moderate | 1 | | | |
| Poor | 0.951 [0.584–1.549] | 0.841 | | |
| Vascular invasion | | | | |
| No | 1 | | | |
| Yes | 1.193 [0.435–3.271] | 0.731 | | |
| MTH1 | | | | |
| Low | 1 | | 1 | |
| High | 1.761 [1.124–2.758] | 0.014* | 1.533 [0.967–2.430] | 0.069 |

Notes:
* $P < 0.05$.
HR, hazard ratio.

Protein levels in human ESCC tissues and cell lines were significantly upregulated, thus, we further explored the potential effects of the MTH1 and NUDT5 protein on the ESCC cell viability. We first established cell models with MTH1 or NUDT5 depletion using two ESCC cell lines (KYSE50 and KYSE70) transfection with either LV-MTH1-RNAi, LV-NUDT5-RNAi and control CON077-(hU6–MCS–Ubiquitin–EGFP–IRES–puromycin) and verified the changes in the expression by Western blotting. As shown in Figs. 2D and 2E, the expression of MTH1 and NUDT5 was significantly reduced. MTH1 and NUDT5 depletion significantly attenuated the cellular proliferation of KYSE50 and KYSE70 cells (Student's *t*-test, $p < 0.05$) according to the CCK-8 proliferation assay (Figs. 2F and 2G). These data suggested that MTH1 and NUDT5 are both positive regulators of ESCC cell proliferation.

**Table 3 Results of univariate and multivariate analyses of the overall survival of NUDT5.**

| | Univariate analysis | | Multivariate analysis | |
| --- | --- | --- | --- | --- |
| | HR (95% CI) | P | HR (95% CI) | P |
| Age (years) | | | | |
| <65 | 1 | | | |
| ≥65 | 0.892 [0.571–1.395] | 0.617 | | |
| Gender | | | | |
| Male | 1 | | | |
| Female | 0.604 [0.348–1.049] | 0.073 | | |
| Lymph node metastasis | | | | |
| No | 1 | | | |
| Yes | 1.377 [0.766–2.473] | 0.285 | | |
| AJCC stage | | | | |
| I + II | 1 | | 1 | |
| III + IV | 3.675 [1.896–7.126] | <0.001* | 1.717 [0.890–3.311] | 0.107 |
| T stage | | | | |
| T1 + T2 | 1 | | | |
| T3 + T4 | 2.615 [1.249–5.477] | 0.011* | 1.630 [0.713–3.726] | 0.247 |
| N stage | | | | |
| N0 | 1 | | 1 | |
| N1 + N2 | 2.065 [1.249–3.293] | 0.002* | 0.760 [0.223–2.594] | 0.661 |
| Differentiation | | | | |
| Well + Moderate | 1 | | | |
| Poor | 0.951 [0.584–1.549] | 0.841 | | |
| Vascular invasion | | | | |
| No | 1 | | | |
| Yes | 1.193 [0.435–3.271] | 0.731 | | |
| NUDT5 | | | | |
| Low | 1 | | 1 | |
| High | 2.084 [1.280–3.393] | 0.003* | 1.751 [1.056–2.903] | 0.030* |

Notes:
* $P < 0.05$.
HR, hazard ratio.

## Depletion of MTH1 and NUDT5 induced cell cycle arrest in ESCC cell lines and affected the expression of cycle-related proteins in KYSE50 and KYSE70 cells

We further performed flow cytometry analysis to investigate whether or not the proliferation inhibition was related to the cell cycle arrest. As shown in Figs. 3A–3L, with MTH1 and NUDT5 depletion, the cells in the G0/G1 phase increased significantly and showed a decline in the S and G2/M phases relative to those in the control groups. This finding suggested that depletion of MTH1 and NUDT5 effectively induced G0/G1 cycle arrest in ESCC cell lines. These results showed that the inhibition of proliferation induced by depleting MTH1 and NUDT5 might occur through cell cycle arrest.

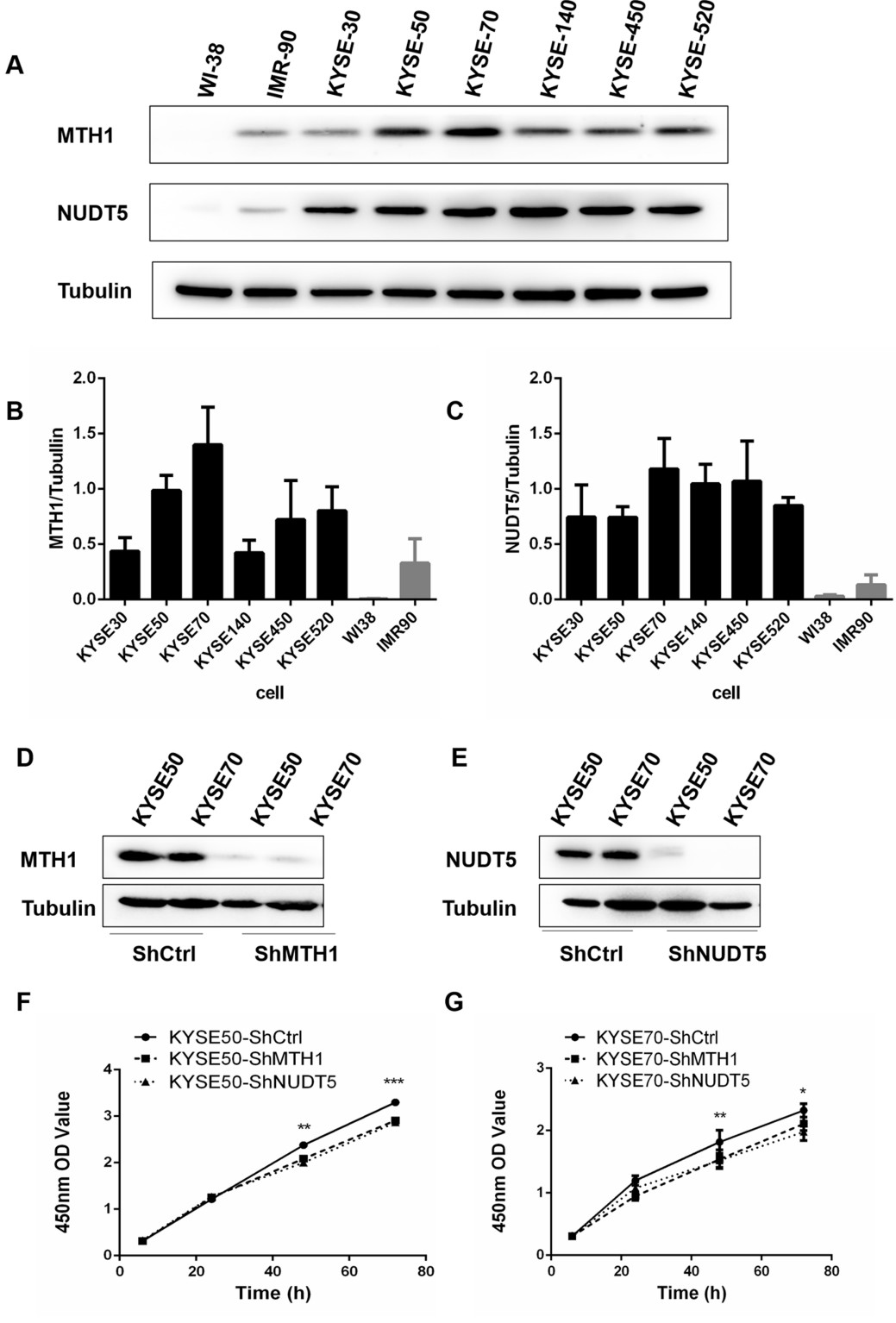

**Figure 2 The expression of the *MTH1* and *NUDT5* genes in human ESCC cell lines. Blocking the expression of MTH1 and NUDT5 depressed the proliferation rate in KYSE50 and KYSE70 cells.** (A) Western blotting of MTH1 and NUDT5 proteins in these ESCC cell lines and WI-38, IMR-90. (B and C) Normalized levels of MTH1 and NUDT5 in six ESCC cell lines, the human normal diploid fibroblast line WI38 and the normal human fibroblast model of senescence IMR90 (Student's *t*-test,

**Figure 2** (continued)

$^*p < 0.05$, compared with WI-38 and IMR-90). (D and E) A Western blotting analysis of MTH1 and NUDT5 in knockdown and control cells. (F and G) Effects of knockdown of MTH1 and NUDT5 protein on the cell proliferation as detected by a CCK-8 assay. All the experiments were repeated three times. $P$ values were calculated by Student's $t$-test, $^*p < 0.05$, $^{**}p < 0.01$, $^{***}p < 0.001$.

To verify the mechanism underlying the involvement of MTH1 and NUDT5 in G0/G1 arrest, we analyzed the expression of cycle-related proteins involved in the regulation of the G1 and S phases by Western blotting. After blocking the expression of MTH1 and NUDT5, the Cyclin D level was decreased (Fig. 3M). Conversely, the levels of p16, p21 and p27 were markedly increased. The changes in cycle-related proteins were more obvious in KYSE50 cells than in KYSE70 cells (Fig. 3M). The results demonstrated that the effects of MTH1 and NUDT5 on G0/G1 phase arrest were related to those proteins specifically targeted in the G1/S transition.

## Depletion of MTH1 and NUDT5 inhibited the migration, invasion and EMT of ESCC cells in vitro

To evaluate the effects of MTH1 and NUDT5 on ESCC cell migration and invasion, Transwell assays were performed. The Transwell migration assays without Matrigel suggested that knockdown of MTH1 or NUDT5 inhibited the migration of KYSE50 and KYSE70 cells. The Transwell invasion assays with Matrigel showed consistent results (Student's $t$-test, $p < 0.05$) (Figs. 4A–4P).

Our data showed that MTH1 and NUDT5 are associated with the migration and invasion of ESCC cells. We therefore explored whether or not EMT of ESCC cells was regulated by MTH1 or NUDT5. We found that a cadherin switch occurred due to the knockdown of MTH1 or NUDT5. Specifically, after the knockdown of MTH1 and NUDT5, E-cadherin expression markedly increased, whereas N-cadherin expression markedly decreased (Fig. 4Q), indicating that EMT was inhibited in the MTH1- and NUDT5-deficient cell lines. Consistent with N-cadherin, the expression of Vimentin (another mesenchymal marker) was also markedly reduced. After blocking MTH1 and NUDT5, the MMP7 expression was found to be markedly reduced (Fig. 4Q). These results suggest that silencing MTH1 and NUDT5 inhibited migration and invasion via the inhibition of EMT.

## The depletion of MTH1 and NUDT5 suppressed ESCC cell aggressiveness via inactivation of the MAPK/MEK/ERK pathway

We next investigated the mechanisms underlying how MTH1 and NUDT5 are involved in the signaling pathway for ESCC cell aggressiveness. We noted that phosphorylation of core members MEK1/2 and ERK1/2 in MAPK pathway was significantly decreased when MTH1 or NUDT5 was knocked down (Fig. 4R). The results in Fig. 4 suggest that MAPK/MEK/ERK activation might underlie the phenotypes induced in ESCC cells by MTH1 and NUDT5.

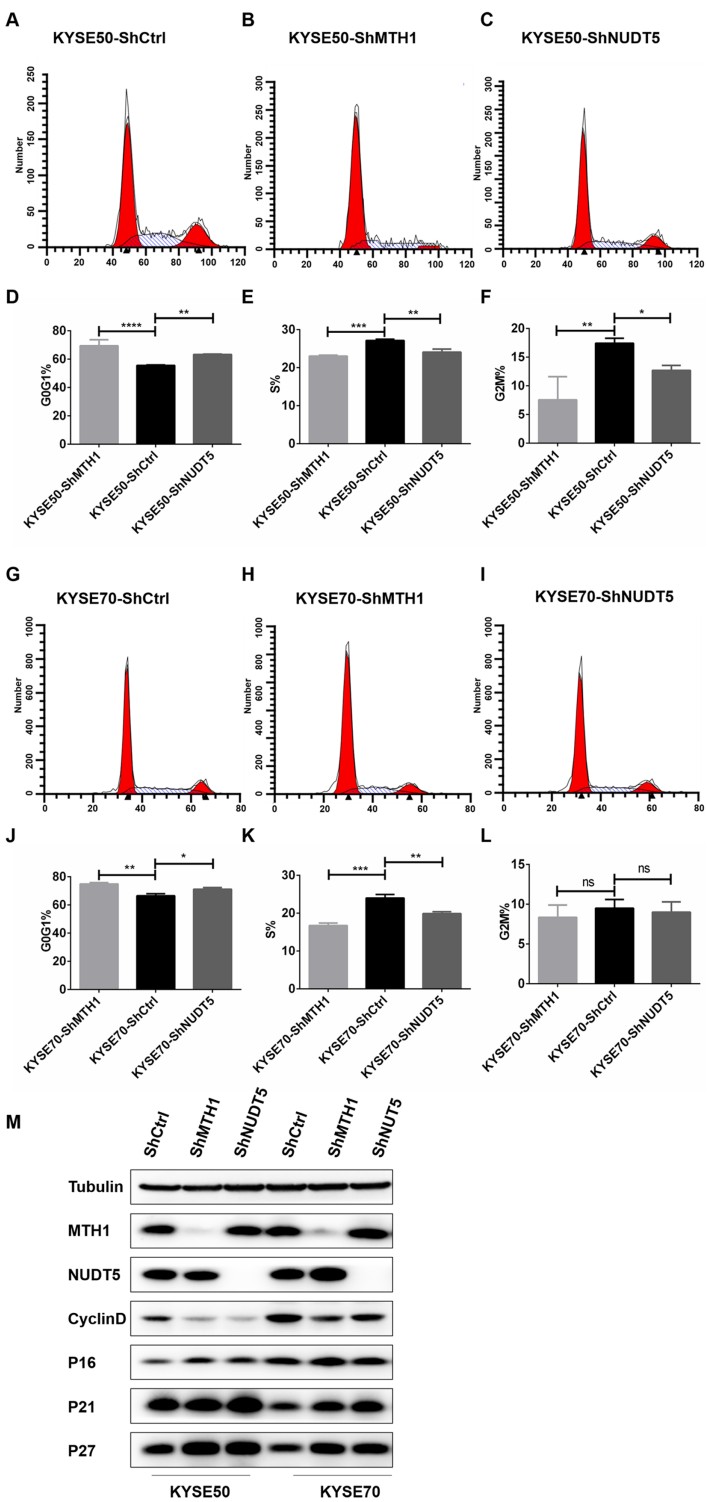

**Figure 3 Blocking the expression of MTH1 and NUDT5 induced cell cycle arrest and affected the expression of cycle-related proteins in the KYSE50 and KYSE70 cell lines.** (A–F) The distribution of the cell cycle (G0/G1, S, G2/M) in the KYSE50 cell line was detected by flow cytometry. The difference between groups in the KYSE50 cell line was analyzed using a two-tailed Student's $t$-test. $^*p < 0.05$ and $^{**}p < 0.01$. Data were expressed as the means ± standard deviation ($^*p < 0.05$, $^{**}p < 0.01$, $^{***}p < 0.001$). (G–L) The distribution of the cell cycle (G0/G1, S, G2/M) was detected by flow-cytometry in the KYSE70

**Figure 3 (continued)**
cell line. The difference between groups in the KYSE70 cell line was analyzed using a two-tailed Student's *t*-test. $^*p < 0.05$ and $^{**}p < 0.01$. Data are expressed as the means ± standard deviation ($^*p < 0.05$, $^{**}p < 0.01$, $^{***}p < 0.001$). (M) Levels of cyclin D, p16, p21 and p27 were detected by Western blotting with tubulin as a control. All the experiments were repeated three times.

## DISCUSSION

NUDIX hydrolases are one of the core nucleotides-metabolizing enzyme families and play critical roles in both health and disease (*Bessman, Frick & O'Handley, 1996*; *McLennan, 2006*; *Mildvan et al., 2005*). NUDIX enzymes are highly expressed in adrenal, endometrial and lung-related cancers, but the opposite is observed in testicular and kidney-related cancers. MTH1 and NUDT5 belong to clusters that are highly expressed in cancer. The high expression of MTH1 was significantly correlated with the tumor pathological stage, lymph node metastasis and prognosis in non-small-cell lung carcinomas (*Kennedy, Pass & Mitchell, 2003*). MTH1 expression is significantly higher in advanced-stage renal cell carcinoma than in early-stage renal cell carcinoma (*Wang et al., 2017b*). In CRC, the levels of MTH1 and NUDT5 were significantly higher in CRC tissues from patients with an advanced AJCC stage and lymph node metastasis and were associated with an extremely poor OS after surgical resection (*Li et al., 2017*). Taken together, these findings indicate that the expression of MTH1 and NUDT5 is correlated with an advanced cancer stage, tumor invasion and a poor prognosis. However, despite the fact that MTH1 and NUDT5 are highly expressed in multiple types of human cancers (*Iida et al., 2001*; *Kennedy et al., 1998*; *Li et al., 2017*; *Okamoto et al., 1996*; *Song et al., 2015*), the clinical importance of MTH1 and NUDT5 in ESCC tissues and the effects of MTH1 and NUDT5 on ESCC cell growth, invasion and metastasis remain unknown.

We revealed that MTH1 and NUDT5 expression in ESCC cell lines and tissues was upregulated, which was consistent with previous studies (*Carreras-Puigvert et al., 2017*). We found that a high expression of MTH1 protein was positively associated with the AJCC and T stages, and high NUDT5 expression was positively correlated with lymph node invasion and the AJCC stage in ESCC patients. A Kaplan–Meier analysis showed that ESCC patients with a high MTH1 or NUDT5 expression had a significantly lower OS than those with a low MTH1 or NUDT5 expression. In addition, the result of multivariate Cox analysis revealed that NUDT5 was an independent prognostic predictor for patients with ESCC. These results suggested that MTH1 and NUDT5 were novel progression and prognostic markers for ESCC and might play a crucial role in the evolution and metastasis of ESCC.

NUDIX enzymes families have been reported to serve as a pro-growth factor in some types of cancers (*Carreras-Puigvert et al., 2017*; *Iyama et al., 2010*; *Oka et al., 2011*). *Gad et al. (2014)* reported that cancer cells required MTH1 activity to avoid the incorporation of oxidized dNTPs. *Oka et al. (2011)* found that, in breast carcinomas, NUDT2 promotes the proliferation of breast carcinoma cells via the HER2 pathways. In the current study, we observed that high MTH1 or NUDT5 expression was associated with a significantly lower OS in patients than the low expression. High expression of

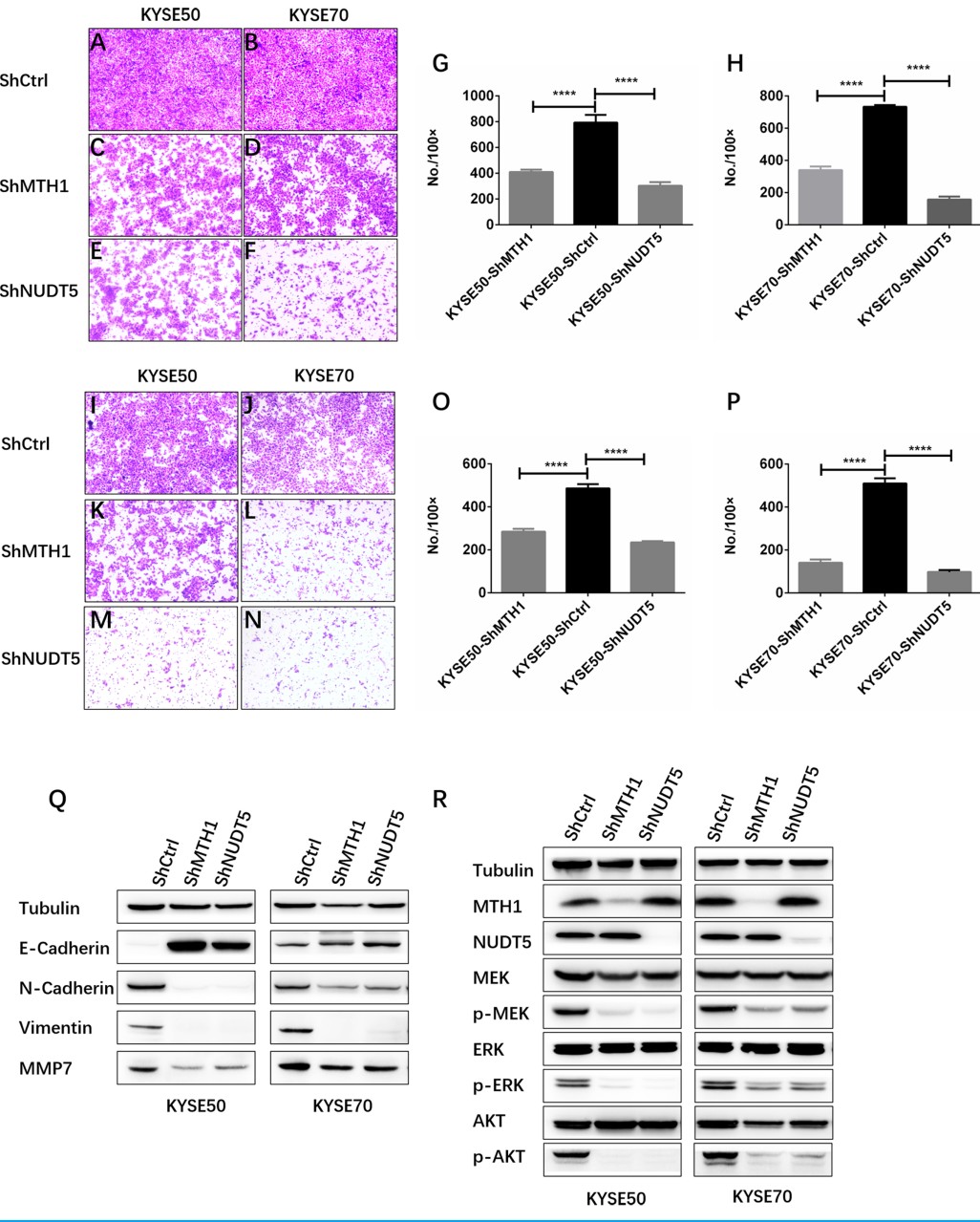

**Figure 4 Depletion of MTH1 and NUDT5 inhibited the migration, invasion and EMT of ESCC cells in vitro.** (A–H) A transwell migration assay and quantification showing the migration ability of shCtrl, shMTH1 and shNUDT5 in KYSE50 or KYSE70 cells. Ten fields were counted per sample and photographs were taken at 48 h. (I–P) A transwell invasion assay and quantification showing the invasion ability of shCtrl, shMTH1 and shNUDT5 in KYSE50 or KYSE70 cells. Ten fields were counted per sample and photographs were taken at 48 h. All data were presented as the means ± SD from three biological replicates (*$p < 0.05$; ** $p < 0.01$). (Q) Western blotting was conducted to determine the protein levels of E-cadherin, N-cadherin, vimentin and MMP7 in MTH1- or NUDT5-knockdown KYSE50 and KYSE70 cells and corresponding vector control cells. (R) Western blotting was conducted to determine the protein levels of MTH1, NUDT5, MEK and ERK and the phosphorylation of MEK or ERK in MTH1- or NUDT5-knockdown KYSE50 and KYSE70 cells. All the experiments were repeated three times.

MTH1 protein was positively associated with the AJCC and T stages and high expression of NUDT5 protein was positively associated with the AJCC stage and lymph node invasion in patients with ESCC. These results suggested that MTH1 and NUDT5 are associated with malignancy of ESCC. We therefore assessed the biological roles of MTH1 and NUDT5 in ESCC cell lines, and the results indicated that proliferation, migration and invasion were significantly suppressed with the depletion of endogenous MTH1 and NUDT5 in vitro. The depletion of NUDT5 significantly decreased the cell viability and significantly delayed the progression of the G1 phase to the S phase (*Zhang et al., 2012*). In our present study, we found that the depletion of MTH1 and NUDT5 significantly inhibited the proliferation of KYSE50 and KYSE70 cells and significantly delayed the cell cycle. After the depletion of MTH1 and NUDT5, the cell cycle progression from the G1 phase was remarkably delayed, indicating that their proliferation capacities had been impaired. The levels of transition proteins, such as p16, p21 and p27, were markedly increased, while the levels of Cyclin D were markedly decreased. This retardation of the cell cycle maybe the reason for the reduction in cell viability observed in MTH1 and NUDT5 knockdown cells.

Evidence suggested that EMT can endow epithelial cells with abilities and functions of migration and invasion that are necessary in the metastasis and invasion processes of various types of cancer and which involve different signal pathways (*Comijn et al., 2001*; *Huber, Kraut & Beug, 2005*; *Mani et al., 2007*). Our results showed that depletion of MTH1 and NUDT5 inhibits the EMT in ESCC. Accumulating evidence showed that the MAPK/MEK/ERK pathway plays a significant role in comprehensive human cancer development (*Dhillon et al., 2007*; *Ding et al., 2016*; *Zhao et al., 2016*). *Wang et al. (2017a)* found that the expression of CTHRC1 in ESCC promoted the migration, invasion and metastasis of tumor cells via the activation of MAPK/MEK/ERK/FRA-1 signaling. In other cancers, such as CRC and breast cancer, MAPK/MEK/ERK signaling was also highly activated with the upregulated of cyclin D1, and proliferation was enhanced (*Ding et al., 2016*; *Zhao et al., 2016*). A previous study showed that AKT and ERK signaling pathways were the cardinal signaling programs mediating EMT in cancer cells. In the present study, we revealed for the first time that MTH1 and NUDT5 mainly exert their effects on the progression of ESCC through the Raf/MEK/ERK pathway. We found that after MTH1 and NUDT5 depletion, the EMT process was inhibited, and mesenchymal markers (MMP7, N-cadherin and vimentin) were downregulated, epithelial marker E-cadherin reflected an increase in expression. We can thus reasonably assume that MTH1 and NUDT5 facilitated the growth, invasion and metastasis of ESCC cells via the regulation of the EMT through the MAPK/MEK/ERK signaling pathway, which in turn promoted ESCC cell metastasis.

In our study, depletion of MTH1 and NUDT5 reduced the expression of p-Akt and p-ERK significantly. AKT, the key molecular of PI3K/AKT signaling pathway plays an important role in proliferation ang cell cycle. Depletion of AKT1/2 results in cell-cycle arrest by upregulating p21 and p27, and downregulating cyclin D and CDK2 (*Ju et al., 2007*; *Santi & Lee, 2011*). Our result consists with these reports. In addition, AKT can inhibit EMT by regulating Twist degradation (*Li et al., 2016a*). Among multiple signaling

pathways which can induce EMT in cancers, MEK/ERK pathway is also a common one. With the depletion of MTH1 and NUDT5, MEK/ERK pathway was significantly inhibited which may alter the expression of several EMT transcription factors such as Zeb and Twist downstream (*Loh et al., 2019*; *Pearlman et al., 2017*). Zeb transcription factors can modulate the expression of epithelial protein. Zeb 1 can be induced by MAPK/MEK/ERK signaling (*Caramel et al., 2013*). Zeb 1 bind to CDH1 which is the promoter domain for E-cadherin and repress the expression of E-cadherin directly (*Zhang, Sun & Ma, 2015*). Zeb 2 is a tumor suppressor which can induce greater cell differentiation (*Zhang, Sun & Ma, 2015*). With the overactivation of MAPK/MEK/ERK signaling, the expression of Zeb 2 switches to Zeb 1 and induces EMT progression (*Zhang, Sun & Ma, 2015*). Transcription factor Twist also can be induced by the activation of MAPK/MEK/ERK signaling pathway (*Koefinger et al., 2011*). In MDCK kidney epithelial cells, cell–cell adhesion and cell polarity were lost with the expression of Twist 1. Meanwhile, proteins which were correlated with EMT were strongly upregulated (*Yang et al., 2004*). EMT occurs downstream of the activation of MAPK/MEK/ERK signaling pathway with the modulation of Twist expression (*Caramel et al., 2013*; *Weiss et al., 2012*). More studies are needed to confirm the target transcription factors of MTH1 and NUDT5.

In addition to the enzymes that act primarily on 8-oxo-dGTP, 8-oxo-GTP and 8-oxo-dGDP, NUDT5 also can acts as a pyrophosphorylase by generating nuclear ATP in the presence of pyrophosphate (PPi) (*Wright et al., 2016*). Nuclear ATP is necessary for chromatin remodeling in key nuclear processes such as DNA replication, proliferation and gene regulation in breast cancer cells (*Wright et al., 2016*). Maybe it's a potential mechanism of which NUDT5 affect the cancer cell cycle. With the overexpression of NUDT5, tumor-relevant proteins MUC1 which was known to upregulate the EMT drivers' expression, was also enriched in breast cancer cell. Our study demonstrates that depletion of MTH1 and NUDT5 could inhibit cell cycle and EMT through MAPK/MEK/ERK and AKT signaling pathway. More studies are needed to confirm the target molecules of MTH1 and NUDT5.

Our results showed that the high expression of MTH1 and NUDT5 were associated with malignancy of ESCC and predicted a poor survival. MTH1 and NUDT5 were novel progression and prognostic markers for ESCC. This is the first time report that the high expression of NUDT5 independently predicted lower OS in patients with ESCC (hazard ratio (HR) 1.751; 95% confidence interval (CI) [1.056–2.903]; $p = 0.030$). Further investigations suggested that upregulated MTH1 and NUDT5 might facilitate the growth, invasion and metastasis of ESCC cells through the MAPK/MEK/ERK signaling pathway. These findings also imply the significance of the NUDIX antioxidant enzyme in the progression of ESCC and will improve our understanding of the mechanisms involved in ESCC.

## CONCLUSIONS

Our results showed that the high expression of MTH1 and NUDT5 were associated with malignancy of ESCC and predicted a poor survival. MTH1 and NUDT5 were novel

progression and prognostic markers for ESCC. Further investigations suggested that upregulated MTH1 and NUDT5 might facilitate the growth, invasion and metastasis of ESCC cells through the MAPK/MEK/ERK signaling pathway. These findings also imply the significance of the NUDIX antioxidant enzyme in the progression of ESCC and will improve our understanding of the mechanisms involved in ESCC.

## ACKNOWLEDGEMENTS

We are grateful to the members of the Institute of Geriatrics of the Ministry of Health for their assistance and advice.

### Funding

This work was supported by the CAMS Innovation Fund for Medical Sciences (No. 2018-I2M-1-002), National Key R&D Program of China (2018YFC2000300) and the National Natural Science Foundation of China (No. 81171028). The funders had no role in study design, data collection and analysis, decision to publish, or preparation of the manuscript.

### Grant Disclosures

The following grant information was disclosed by the authors:
CAMS Innovation Fund for Medical Sciences: 2018-I2M-1-002.
National Key R&D Program of China: 2018YFC2000300.
National Natural Science Foundation of China: 81171028.

### Competing Interests

The authors declare that they have no competing interests.

### Author Contributions

- Jing-Jing Wang conceived and designed the experiments, performed the experiments, analyzed the data, prepared figures and/or tables, authored or reviewed drafts of the paper, and approved the final draft.
- Teng-Hui Liu performed the experiments, authored or reviewed drafts of the paper, and approved the final draft.
- Jin Li conceived and designed the experiments, performed the experiments, authored or reviewed drafts of the paper, and approved the final draft.
- Dan-Ni Li analyzed the data, authored or reviewed drafts of the paper, and approved the final draft.
- Xin-Yuan Tian analyzed the data, authored or reviewed drafts of the paper, and approved the final draft.
- Qiu-Geng Ouyang performed the experiments, authored or reviewed drafts of the paper, and approved the final draft.
- Jian-Ping Cai conceived and designed the experiments, prepared figures and/or tables, authored or reviewed drafts of the paper, and approved the final draft.

## DNA Deposition

The following information was supplied regarding the deposition of DNA sequences:

RNA sequences of MTH1 described here are available at GenBank: NM_002452.3 (401-419).

RNA sequences of NUDT5 described here are available at GenBank: NM_014142.3 (525-542).

## Data Availability

The raw data, the commercial tissue microarrays, and other raw measurements are available in the Supplemental Files.

## Supplemental Information

Supplemental information for this article can be found online at http://dx.doi.org/10.7717/peerj.9195#supplemental-information.

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
