# Peer review of "The high expression of MTH1 and NUDT5 predict a poor survival and are associated with malignancy of esophageal squamous cell carcinoma"

_PeerJ, doi:10.7717/peerj.9195_

## Round 0.1 · original submission · Major Revisions

Your manuscript has been reviewed by two experts in the field. As you will find from their comments below, both of them basically admit the significance of this work but raise a number of (mostly minor) points. For your information, there has been the third reviewer but I decided to make the decision without waiting for his/her comments for your time convenience. I would like to add, however, that the third reviewer points out that many (?) of the references from #3 to #18 seem inappropriate in personal communication with me. Please check them, again. Looking forward to receiving your revised manuscript.

Reviewer 1 ·

Basic reporting

Wang et al. analyzed the prognostic value and in-vitro function of MTH1 and NUDT5 in esophageal squamous carcinoma (ESCC). Several points should be corrected.
1. Abstract: “Protein expression were in ESCC cell lines were…” should be corrected to “Protein expression in ESCC cell lines were…”.
2. Table 1: Alternating use of Chi-squared test and Fisher test looks arbitrary and is not favorable. Please use the same test throughout Table 1. Fisher test would be better. If the authors want use Chi-squared test, please explain the rationale.
3. Table 2 and 3: HRs for vascular invasion are different between Table 2 and 3. Why?
4. Table 2. The bottom row may be MTH1.

Experimental design

Experiments are well designed, and the methods are well described.

Minor point.
Line 129: The catalog numbers of antibodies for MTH1 and NUDT5 should be reported for reproducibility of the study.

Validity of the findings

It would be helpful if the authors discuss potential mechanism of how MTH/NUDT5 silencing causes cell cycle arrest and EMT inhibition.

Reviewer 2 ·

Basic reporting

Wang et al have a clear scientific aim and uses a sound methodological approach, using human tissue from the disease of interest, human cell lines, standard methods such as FACS, IHC, western blot, knockdown experiments et c to proof their hypothesis. The introduction and background are relevant, and a professional language is used. The literature references are relevant. I miss some explanation about the various stages of the disease that are then refered to in the result part (i.e. AJCC, T and N stage). It could useful to mention something about the disease stages in the background.

Figures:
1. Figure 1A-I need some more clarifications. It states that A-H is representative IHC for MTH1 and NUDT5 expression in ESCC specimens. Is it from two different patients, where A and B comes from the same patient and C and D are adjacent normal vs tumor specimens from another patient? Or is it four different patients? Please clarify. Figure 1I- is this data mean from 99 patients from IHC data or western blot? Please clarify in figure legend how many patients and method used to analyse the protein expression. The raw data for Figure A-I is missing.
2. Figure 2. Please add number of independent experiments in figure legend. Raw data from figure 2F-G is missing.
3. Figure 3. Please add number of independent experiments performed in the figure legend. Please improve font on FACS figures A, C. Very difficult to read legends.
4. Figure 4. Please add the number of independent experiments performed in the figure legend. Looking at the raw data for fig 4A-B, and reading the text, it indicates that it is only one independent experiment, but 10 field counted per sample. Please clarify. If n=1, it should be repeated to secure reproducibility and conclusion. Similarly, how many times have the western blot been performed in fig 4C-D? Is this a representative blot out of x experiments? If yes, please add the other blot as supplementary figure. If only one WB experiment, it should be repeated to make the conclusions stronger.

Minor lay-out/text/reference comments:
5.. Lacking reference to figures in the result text. For instance, row 219-220 a result statement but no reference to data. Same for row 221 and row 226.
6.. Row 78: …”induced by ROS is 8-Oxo-7,…” should be …”induced by ROS is 8-oxo-7,…”
7. Row 86: reference 13 should be formatted correctly
8.. Row 128, reference 20 should be formatted correctly
9.. Row 133: reference 20 should be formatted correctly
10.. Row 174: ….duration at 3 x 103 cells per”…, is it really 3 x 103 or is it 3 x 10000 ?
11.. Row 185, 186: ….”70% alcohol..”, please specify what alcohol, assume it is ethanol
12. Row 197: …”8x104 cells”.., or is it 8 x 100000 ?

Experimental design

Row 121-124: Please clarify what you mean with paired ESCC specimens vs tumor tissues singly? (I guess it means 81 paired tumor and adjacent normal specimens and 18 ESCC tumor tissues singly).

Please add in the material and method section the ethical permit number showing that you are allowed to use the human tumor tissues and clinical data .

Validity of the findings

Missing some information on independent replicates of experiments (see above).
The authors should show that it has been repeated (n=3) and the same results are obtained.

The conclusions from figure 2F-G on row 351 : "..the depletion of MTH1 and NUDt5 remarkebly inhibited the proliferation of ..." is a bit strong when looking at the data. Suggest to change "remarkably" to significantly.

---

## Round 0.2 · accepted · Accept

Your revised manuscript has been reviewed by the same two referees. Fortunately, both of them now recommend its acceptance to me. Thus, I am happy to make the decision of its acceptance.

Reviewer 1 ·

Basic reporting

The authors correctly revised the basic reporting of this munuscript.

Experimental design

I see no problem in the Experimental design of this manuscript.

Validity of the findings

I have no concern about the Validity of the findings of this manuscript.